# Uncovering the BIN1-SH3 interactome underpinning centronuclear myopathy

Boglarka Zambo[1], Evelina Edelweiss[2], Bastien Morlet[2], Luc Negroni[2], Matyas Pajkos[3], Zsuzsanna Dosztanyi[3], Soren Ostergaard[4], Gilles Trave[1]*, Jocelyn Laporte[2]*, Gergo Gogl[1]*

[1]Equipe Labellisee Ligue 2015, Departement de Biologie Structurale Integrative, Institut de Genetique et de Biologie Moleculaire et Cellulaire (IGBMC), INSERM U1258/CNRS UMR 7104/Universite de Strasbourg, Illkirch, France; [2]Institut de Genetique et de Biologie Moléculaire et Cellulaire (IGBMC), INSERM U1258/CNRS UMR 7104/Université de Strasbourg, Illkirch, France; [3]Department of Biochemistry, ELTE Eötvös Loránd University, Budapest, Hungary; [4]Novo Nordisk A/S, Global Research Technologies, Novo Nordisk Research Park, Maaloev, Denmark

**Abstract** Truncation of the protein-protein interaction SH3 domain of the membrane remodeling Bridging Integrator 1 (BIN1, Amphiphysin 2) protein leads to centronuclear myopathy. Here, we assessed the impact of a set of naturally observed, previously uncharacterized BIN1 SH3 domain variants using conventional *in vitro* and cell-based assays monitoring the BIN1 interaction with dynamin 2 (DNM2) and identified potentially harmful ones that can be also tentatively connected to neuromuscular disorders. However, SH3 domains are typically promiscuous and it is expected that other, so far unknown partners of BIN1 exist besides DNM2, that also participate in the development of centronuclear myopathy. In order to shed light on these other relevant interaction partners and to get a holistic picture of the pathomechanism behind BIN1 SH3 domain variants, we used affinity interactomics. We identified hundreds of new BIN1 interaction partners proteome-wide, among which many appear to participate in cell division, suggesting a critical role of BIN1 in the regulation of mitosis. Finally, we show that the identified BIN1 mutations indeed cause proteome-wide affinity perturbation, signifying the importance of employing unbiased affinity interactomic approaches.

**\*For correspondence:**
travegi@igbmc.fr (GT);
jocelyn@igbmc.fr (JL);
goglg@igbmc.fr (GG)

**Competing interest:** The authors declare that no competing interests exist.

## eLife assessment

This work describes a novel affinity interactomics approach that allows investigators to identify networks of protein-protein interactions in cells. The **important** findings presented here describe the application of this technique to the SH3 domain of the membrane remodeling Bridging Integrator 1 (BIN1), the truncation of which leads to centronuclear myopathy. The authors present **solid** evidence that BIN1 SH3 engages with an unexpectedly high number of cellular proteins, many of which are linked to skeletal muscle disease, and evidence is presented to suggest that BIN1 may play a role in mitosis creating the potential for new avenues in drug development efforts. Some of the findings, however, remain rather preliminary, lack sufficient replicates and may require additional experiments to definitively support the conclusions.

## Introduction

Bridging Integrator 1 (BIN1), also known as Amphiphysin 2 (AMPH2), is a ubiquitously expressed membrane remodeling protein. It contains an N-terminal BAR domain required for membrane binding

**Figure 1.** Involvement of BIN1 in membrane remodeling and a compendium of known BIN1 interaction partners. (**A**) Models of vesicle and T-tubule formation in the context of BIN1 and DNM2. BIN1 interplays in both processes, through its membrane bending/tubulating BAR domain and its SH3 domain. In clathrin-, or caveolin-coated vesicle formation, as well as during the formation of recycling endosomes, the recruitment of DNM2 by BIN1 is critical for vesicle scission. During T-tubule formation, DNM2 is also recruited, but in this case less scission occurs. (**B**) Schematic illustration of binding of PRMs to SH3 domains. Due to the twofold pseudo-symmetry of PPII helices, class 1 and class 2 PxxP motifs bind in different orientations to SH3 domains (*Lim et al., 1994*). (**C**) Known interaction partners of BIN1 identified by high-throughput qualitative interactomic studies and the experimental overlap between the different sources (*Cho et al., 2022*; *Ellis et al., 2012*; *Huttlin et al., 2021*; *Luck et al., 2020*). Note that the known SH3-domain mediated interaction partners, that were studied by low-throughput methods, were only detected on a few occasions (DNM2, MYC, RIN3, marked in orange), or not detected at all (TAU/MAPT, CAVIN4).

— named after BIN1, AMPH, and RVS167 — and a C-terminal SRC Homology 3 (SH3) domain, required for partner recruitment (*Owen et al., 1998*; *Peter et al., 2004*; *Prokic et al., 2014*). BAR domain deposition on membrane surfaces causes membrane curvature and BIN1-mediated membrane remodeling was found to be critical in the formation of various endomembrane structures, such as clathrin- or caveolin-coated vesicles, recycling endosomes, as well as tubular invaginations of the plasma membrane in muscle cells, known as T-tubules (*Lee et al., 2002*; *Ramjaun and McPherson, 1998*; *Razzaq et al., 2001*). A well-studied role of the BIN1 SH3 domain is to recruit Dynamin 2 (DNM2) to curved membranes, whose local oligomerization appears to be critical in both membrane fission during vesicle scission and in the formation of T-tubules in muscle cells (*Chin et al., 2015*; *Cowling et al., 2017*; *David et al., 1996*; *Fujise et al., 2022*; *Volchuk et al., 1998*; *Figure 1A*). Both BIN1 and DNM2 are implicated in centronuclear myopathy (CNM): mutations of DNM2 were found to lead to autosomal dominant CNM and mutations of BIN1 were found to lead to autosomal recessive CNM (*Bitoun et al., 2005*; *Gómez-Oca et al., 2022*; *Nicot et al., 2007*; *Rossi et al., 2022*). Disease-associated mutations of BIN1 can occur at several positions. Pathological BIN1 mutations located in its BAR domain prevent its membrane-remodeling function, and almost completely abolish BIN1-related cellular mechanisms (*Nicot et al., 2007*). Importantly, rare truncations of the SH3 domain of BIN1 (caused by early stop codons such as Q573*, or K575*) also result in CNM (*Laiman et al., 2023*; *Nicot et al., 2007*). These BIN1 variants – whose BAR domains are intact – maintain their membrane remodeling activities, such as creating tubular membrane structures, yet are unable to recruit DNM2. In addition, a frameshift mutation causing a mostly hydrophobic 52-residue extension of the BIN1 SH3 domain was found to cause autosomal dominant CNM (*Böhm et al., 2014*). These observations suggest the critical role of the SH3 domain of BIN1 and its mediated protein-protein interactions in CNM. Exogenous expression of BIN1 is a promising therapeutic approach to treat different genetic forms of CNM, reinforcing the importance of characterizing the interactions of BIN1 and their functional consequences (*Lionello et al., 2022*; *Lionello et al., 2019*).

SH3 domains recognize Proline-rich motifs (PRMs), most typically basic PxxP motifs (*Lim et al., 1994*; *Figure 1B*). DNM2 contains an extensive Proline-rich region (PRR) in its C-terminal tail including a series of putative PRMs that are thought to interact with the SH3 domain of BIN1 (*Grabs et al., 1997*). Consequently, the widely accepted view is that disruption of cellular BIN1-DNM2 complex leads to CNM. However, SH3 domains are typically highly promiscuous and can bind to hundreds of

partners, similarly to other protein-protein interaction domain families (*Gogl et al., 2022*; *Wu et al., 2007*). So far, only a handful of other SH3-mediated interactions of BIN1 were identified, such as MYC, TAU, RIN3, and Caveolae-associated protein 4 (CAVIN4) (*Andresen et al., 2012*; *Kajiho et al., 2003*; *Lasorsa et al., 2018*; *Lo et al., 2021*; *Malki et al., 2017*; *Pineda-Lucena et al., 2005*). Viral proteins were also found to interact with the BIN1 SH3 domain (*Nanda et al., 2006*; *Tossavainen et al., 2016*). In addition to these interactions of the BIN1 SH3 domain identified with low-throughput approaches, the interactome of full-length (FL) BIN1 was also screened in multiple high-throughput interactomic studies (*Cho et al., 2022*; *Ellis et al., 2012*; *Huttlin et al., 2021*; *Luck et al., 2020*). However, the results of these studies overlapped poorly (*Figure 1C*). Therefore, it remains an open question whether BIN1 has other interactions besides DNM2, how important these partners are, and how BIN1 variants associated with CNM affect this interactome.

Here, we investigate the biophysical consequences of several previously uncharacterized natural BIN1 SH3 domain variants. We show that tentatively pathological variants are affecting not only the previously well-characterized interaction with DNM2, but also hundreds of other previously unknown BIN1 interactions. We showed this by charting an unbiased affinity interactomic map of the SH3 domain of BIN1 using a top-down affinity interactomic strategy, exploiting the full potential of our innovative experimental approaches (*Gogl et al., 2022*; *Zambo et al., 2022*). Using our recently developed native holdup approach we investigated the binding of BIN1 SH3 to nearly 7000 FL proteins from total cell extracts, out of which we could quantify apparent dissociation constants for ca. 200 interaction partners. Then, we identified and synthesized all putative PRMs found in their sequence (448 PxxP motifs), and systematically measured their binding affinities with the SH3 domain of BIN1 in order to reveal the site-specific molecular mechanisms behind the observed BIN1 interactions. Analyzing the identified partners that interact with BIN1 through well-defined PxxP motifs, we have found that many of them are involved in cell division and thus we concluded that BIN1 could contribute to the regulation of mitosis through specific partners such as PRC1. Finally, by exploiting this peptide library that includes all PRMs that are found in all relevant BIN1-partners, we could precisely quantify the impact of a set of natural missense variants on the site-specific affinity interactome of the SH3 domain of BIN1.

## Results

### The impact of missense BIN1 SH3 variants on DNM2-related phenotypes

We used the holdup approach to study the consequences of rare BIN1 variants located in the SH3 domain on DNM2 binding. Holdup is an established method to quantify equilibrium binding constants between resin-immobilized bait and in-solution analyte molecules (*Charbonnier et al., 2006*; *Gogl et al., 2022*; *Vincentelli et al., 2015*). First, a purified bait molecule or a control compound is immobilized on resin at a sufficient quantity to reach resin-saturating conditions. Then, this bait-saturated and control resin stock are mixed with a dilute analyte solution forming a thick resin slurry where the bait concentration reaches high concentrations. After a brief incubation, the binding equilibrium is reached where the free and bound prey molecules are separated in different phases. By rapidly separating these phases it is possible to measure how much of the prey molecule is depleted from the supernate. The main advantage of holdup over conventional pulldown-based approaches, such as immunoprecipitation, is that it captures the undisturbed binding equilibrium allowing the determination of steady-state binding constants, instead of measuring the enrichment of bound prey on the resin after washing steps that only allows qualitative assessment of binding (*Charbonnier et al., 2006*). The measured relative prey depletion, often referred to as binding intensities (BI), can be converted to equilibrium dissociation constants (*Delalande et al., 2022*; *Gogl et al., 2020*; *Gógl et al., 2019*; *Vincentelli et al., 2015*). For this assay, we used streptavidin resin saturated with either biotin (control) or a synthetic biotinylated peptide derived from the C-terminal PRR region of DNM2 (residues 823–860) as a bait. As a prey, we used recombinant BIN1 SH3 domains. We selected one common, likely benign (T532M), and eight rare variants with unknown clinical significance (Y531S, D537V, Q540H, P551L, V566M, R581C, V583I, and F584S) of the BIN1 SH3 domain from genomic databases, such as dbSNP, ClinVar, or gnomAD (*Karczewski et al., 2020*; *Landrum et al., 2020*; *Sherry et al., 2001*). We produced these SH3 domain variants recombinantly. The holdup assay

revealed that the variants Y531S, D537V, and F584S do not detectably bind to the PRR of DNM2, while the other variants display affinities similar to the WT BIN1. In agreement with these findings, artificial point mutations were used to map the DNM2 binding interface on BIN1 in the past and two studied mutations, Y531F and D537A coinciding with Y531S and D537V natural variants, were also found to cause a marked loss of DNM2 binding activity (*Owen et al., 1998*).

Clinically important, CNM-causing BIN1 variants were previously found to be unable to recruit DNM2 to BIN1-induced membrane invaginations that resemble T-tubules (*Nicot et al., 2007*). To test if the Y531S, D537V, and F584S variants can also reproduce the same phenotype, we co-transfected Cos-1 cells with GFP-BIN1 (full-length, isoform 8) and Myc-DNM2 (full-length) as described previously (*Fujise et al., 2021*; *Lionello et al., 2022*; *Figure 2B–C*, *Figure 2—figure supplement 1*). As a control, we also performed the membrane tubulation assay with WT BIN1 and with the likely benign Q540H variant. As expected, all tested BIN1 variants were capable of promoting tubular endomembrane structures since this process is mediated by the BAR domain of BIN1 and not by its SH3 domain. Furthermore, the WT and the likely benign Q540H variant were capable of efficiently anchoring DNM2 to these membrane tubules. Unexpectedly, the F584S variant, which showed no binding to DNM2 *in vitro* could also recruit DNM2 in cells. However, the tubules formed by this variant appeared to be less polarized and organized than the WT. In contrast, the Y531S and D537V variants showed markedly reduced ability to recruit DNM2, the same cellular phenotype as previously observed in the case of BIN1 variants that cause CNM (*Nicot et al., 2007*; *Figure 2C–E*).

## Deciphering the intrinsic affinity interactome of selected SH3 domains using native holdup

So far, using the DNM2 PRR as a peptide bait in an *in vitro* holdup assay, we identified three previously uncharacterized BIN1 variants that display impaired DNM2 binding, which also resulted in certain altered cellular phenotypes. Yet, these results provide a limited insight into the role of the SH3 domain of BIN1 in CNM as this investigation was restricted only to its DNM2 interaction. To explore the deeper molecular mechanisms underlying BIN1-related CNM, we decided to use an unbiased affinity interactomic approach to obtain a more complete picture of the quantitative interactome of the SH3 domain of BIN1 in comparison with other SH3 domains, which may help shed light on the molecular network aberrations and new relevant protein partners underlying myopathies.

A recent version of the holdup approach, called native holdup (nHU), uses dilute cell extracts as analyte instead of a purified protein prey, providing estimates of equilibrium dissociation constants for thousands of endogenous FL proteins from a single experiment (*Zambo et al., 2022*). This version of the holdup assay builds on the assumption that the binding affinities of even thousands of prey molecules, all present in a dilute cell extract, can be precisely and simultaneously measured as even their cumulative bound quantities is negligible compared to the large amount of resin-immobilized exogenous bait. To decide if nHU could be used to capture interactions of the SH3 domain of BIN1, we comparatively measured the affinities between BIN1 and DNM2 using recombinant BIN1 SH3 domain (BIN1_SH3) as a bait and either catalytically active recombinant DNM2 (purified protein) or endogenous DNM2 found in total myoblast extract as a prey and monitored DNM2 binding using Western blot (*Figure 3A*, *Figure 3—figure supplements 1–2*). We found that purified DNM2 interacts nearly identically to endogenous DNM2 found in myoblast extract with an apparent dissociation constant of 100 nM (p$K_{app}$ = 7). Interestingly, these titration experiments also revealed that DNM2 displays partial binding activity, i.e. not the entire DNM2 population is capable of interacting with the SH3 domain of BIN1. The holdup experiment using purified DNM2 and nHU experiment using endogenous DNM2 found in myoblast extracts produced nearly identical results. Thus, we concluded that nHU could be used to reliably capture interactions of SH3 domains and determine their steady-state binding constants.

We used this approach coupled with mass spectrometry (MS) to estimate the steady-state dissociation constants of all complexes formed between the BIN1 SH3 domain (BIN1_SH3) and all detectable FL protein from total Jurkat extracts. We performed single-point nHU experiments at 10 µM estimated bait concentration, quantified the prey depletion with label-free quantitative MS, and converted the obtained fraction bound (binding intensity, BI) values to apparent equilibrium dissociation constants using a simple bimolecular binding model (hyperbolic formula) (*Figure 3*, *Supplementary file 1*). In our assay, we assayed the binding of 6,357 FL proteins, out of which 188 showed statistically

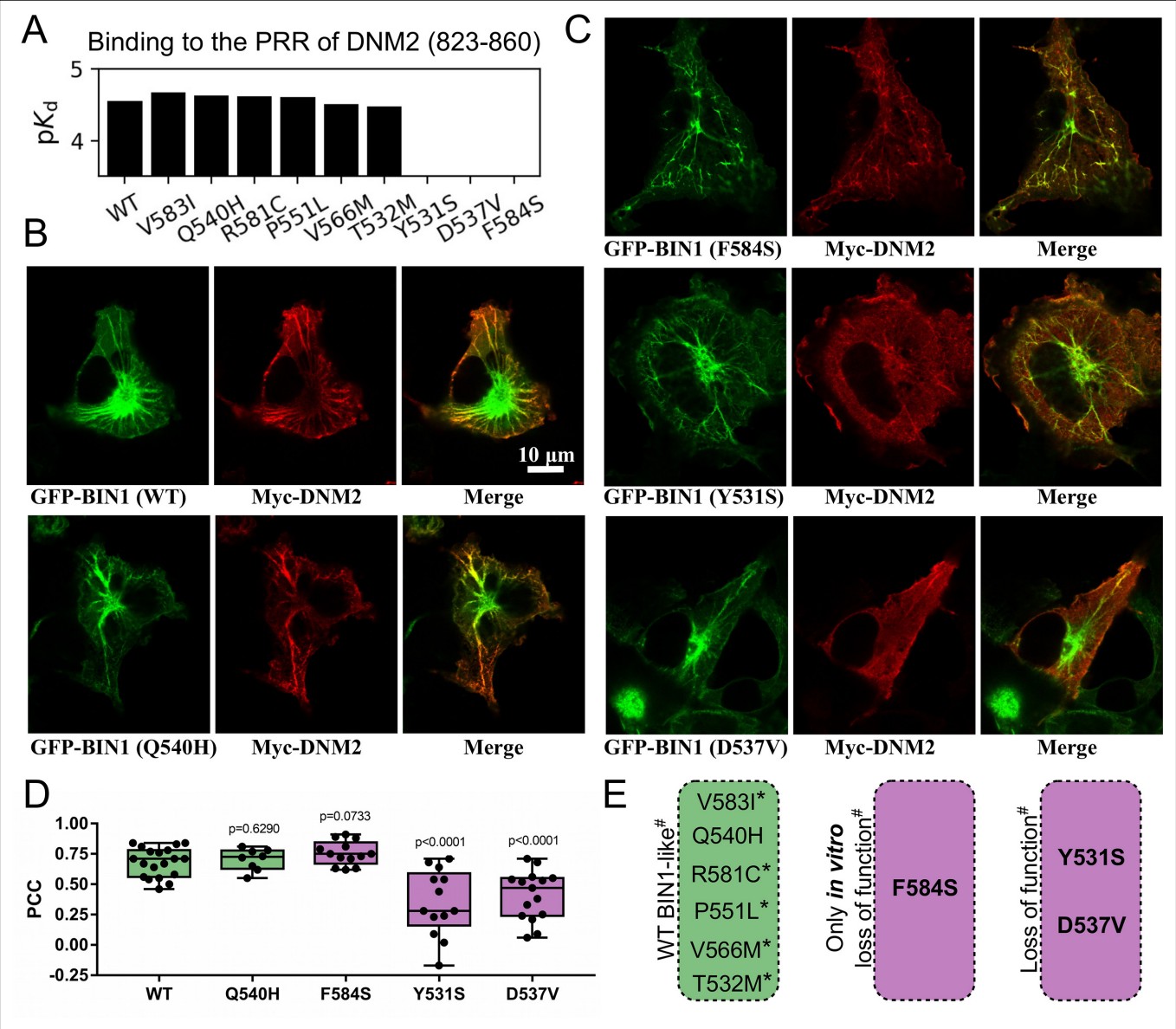

**Figure 2.** Several BIN1 variants of unknown clinical significance have a strong impact on the binding of DNM2 and display altered cellular phenotype. (**A**) Measured affinities of the PRR of DNM2 against a set of natural BIN1_SH3 variants. Most variants interact with DNM2 with similar affinities, but Y531S, D537V and F584S variants disrupt this interaction. Affinities are expressed as negative logarithm of dissociation constants, i.e. p$K_d$ 4 equals to 100 μM $K_d$. (**B**) Membrane tubulation assay performed with WT BIN1 and DNM2, as well as Q540H variant which binds DNM2 with the same affinity as WT BIN1. (**C**) Membrane tubulation assay performed with the variants displaying decreased affinities to DNM2. Cos-1 cells were transfected with GFP-BIN1 and Myc-DNM2. The effect of F584S seems to be apparently rescued in the context of FL BIN1, but both Y531S and D537V variants are unable to efficiently recruit DNM2 to membrane tubules in cells. (**D**) Statistical analysis of single-cell co-localization experiments between the BIN1 variants and DNM2 (n[WT]=19, n[Q540H]=8, n[F584S]=13, n[Y531S]=13, n[D537V]=15). P values were calculated between Pearson correlation coefficients (PCC) of WT and missense variants using a two-tailed unpaired Student's T-test. Box plots indicate the median and upper and lower quartiles, and whiskers label the minimal and maximal measured PCC values. Individual data points representing measurements of single cells are also indicated. (**E**) A summary of the effects of the BIN1 variants. Asterisk indicates that the variants were only tested *in vitro*, and # indicates that the effects were measured based on the BIN1-DNM2 interaction phenotype. See *Figure 2—figure supplement 1* for additional images.

The online version of this article includes the following figure supplement(s) for figure 2:

**Figure supplement 1.** Additional images of membrane tubulation assay performed with BIN1 variants.

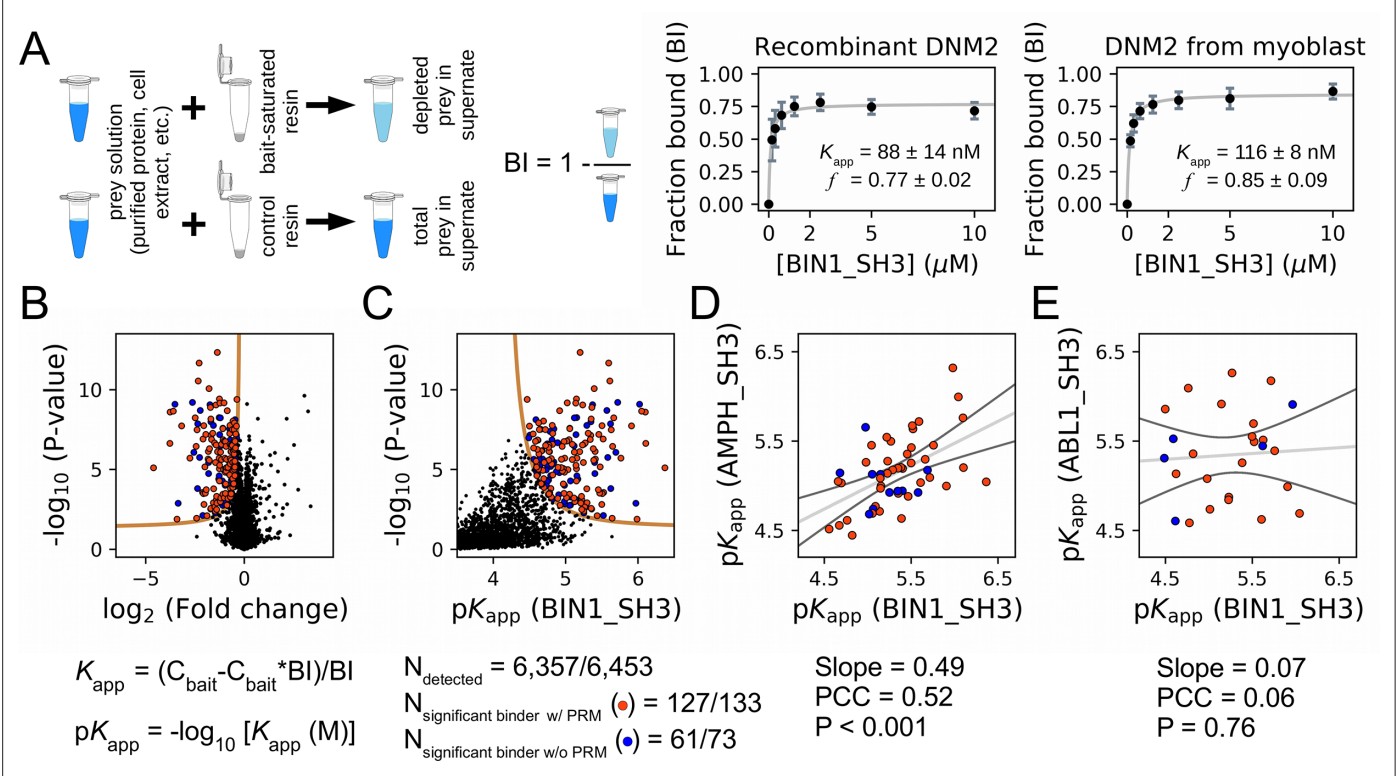

**Figure 3.** Affinity measurements between the SH3 domain of BIN1 and full-length proteins from cell extracts using nHU-MS. (**A**) Outline of the holdup assay and benchmarking of nHU. Holdup is a simple tool to measure the fraction bound of prey molecules. This prey solution can be either a single purified protein, or a complex mixture of molecules and the prey depletion can be monitored with a multitude of analytical approches, such as western blot, or mass spectrometry. Titration holdup experiments were used to further characterize the interactions of BIN1_SH3 and full-length DNM2 using either recombinant, purified DNM2, or total myoblast extract containing endogenous DNM2 as a prey. The two experiments show nearly identical binding affinity and partial activity. (The average and standard deviation of three nHU western blot experiments are shown.) (**B**) Results of single point nHU-MS experiments carried out with the SH3 domain of BIN1 and total Jurkat extracts. Interaction partners above the significance threshold (tan line) are colored in orange if putative PRMs were identified in their sequence and blue if not. (**C**) Measured depletion values were converted to affinities using the functions indicated below panel B, assuming a simple binding mechanism and 10 μM estimated bait concentration. The number of unique affinity measurements performed and the identified BIN1 interaction partners found in a single experiment/in all measurements are indicated below panel C. (**D, E**) We also performed nHU-MS experiments with a set of closely or distantly related SH3 domains and compared their affinity profiles with BIN1. This way, we could quantify that related SH3-domains, for example the one found in AMPH, show similarities in their affinity interactomes, displaying statistically significant correlation between the measured affinity constants. In contrast, unrelated SH3 domains, such as the one found in ABL1, bind targets with dissimilar affinities. A linear fit (grey line) and a 95% confidence band (black line) is shown on all affinity comparisons. The statistical significance of correlation was determined by two-tailed, unpaired T-test. See *Figure 3—figure supplements 1–3* and *Supplementary file 1* for further details. Mass spectrometry experiments were performed with three injection replicates.

The online version of this article includes the following source data and figure supplement(s) for figure 3:

**Figure supplement 1.** Quality control of purified DNM2.

**Figure supplement 1—source data 1.** Original SDS-PAGE for panel A.

**Figure supplement 2.** Raw results of the titration nHU and titration HU experiments.

**Figure supplement 2—source data 1.** Original western blot image (overlayed with colorimetric image) of Membrane 1.

**Figure supplement 2—source data 2.** Original western blot image (overlayed with colorimetric image) of Membrane 2.

**Figure supplement 2—source data 3.** Original western blot image (overlayed with colorimetric image) of Membrane 3.

**Figure supplement 2—source data 4.** Original western blot image (overlayed with colorimetric image) of Membrane 4, DNM2.

**Figure supplement 2—source data 5.** Original western blot image (overlayed with colorimetric image) of Membrane 4, GAPDH.

**Figure supplement 2—source data 6.** Original western blot image (overlayed with colorimetric image) of Membrane 5, DNM2.

**Figure supplement 2—source data 7.** Original western blot image (overlayed with colorimetric image) of Membrane 5, GAPDH.

**Figure supplement 2—source data 8.** Original western blot image (overlayed with colorimetric image) of Membrane 6, DNM2.

*Figure 3 continued*

**Figure supplement 2—source data 9.** Original western blot image (overlayed with colorimetric image) of Membrane 6, GAPDH.

**Figure supplement 2—source data 10.** Original western blot image (overlayed with colorimetric image) of Membrane 7, PRC1.

**Figure supplement 2—source data 11.** Original western blot image (overlayed with colorimetric image) of Membrane 7, GAPDH.

**Figure supplement 2—source data 12.** Original Western blot image (overlayed with colorimetric image) of Membrane 8, PRC1.

**Figure supplement 2—source data 13.** Original western blot image (overlayed with colorimetric image) of Membrane 8, GAPDH.

**Figure supplement 2—source data 14.** Original western blot image (overlayed with colorimetric image) of Membrane 9, PRC1.

**Figure supplement 2—source data 15.** Original western blot image (overlayed with colorimetric image) of Membrane 9, GAPDH.

**Figure supplement 3.** Additional results of nHU-MS experiments.

significant binding to BIN1_SH3 displaying apparent dissociation constants in the range of 0.5–34 µM (corresponding to 6.4–4.5 p$K_{app}$ values). In an ideal nHU experiment, any protein that shows specific depletion in the presence of bait-saturated resin is considered a binder. In reality, the sensitivity of the proteomic measurement limits the accurate detection of all binders, as mass spectrometry cannot quantify all proteins equally well. This is particularly true for low-affinity binding partners, where very small differences need to be quantified and proportionally higher detection noise results in lower significance values. These proteins often remain below our strict statistical threshold, despite the fact that their measured depletion ratio can be close to their theoretical depletion ratio, set by their intrinsic affinities and the conditions of the holdup assay. Therefore, it is important to keep in mind that any proteins displaying depletion may also be true interaction partners of the bait, regardless of their statistical significance. Once more evidence emerges about these interactions in the future, their measured depletion values can be reconsidered and their affinities can be further evaluated.

In a nHU experiment, as we determine an intrinsic parameter of molecular interactions, we expect to obtain the same depletion ratio in biological replicates. To determine the analytical error, which we consider to be more significant than the biological variability, we decided that it is sufficient to perform technical replicates instead of biological replicates. To provide additional validations of the first nHU experiment, we repeated this experiment under identical conditions with a different batch of purified BIN1_SH3 bait and a newly prepared Jurkat extract. The analytics of this second measurement was performed on a less capable, yet highly robust mass spectrometer. Consequently, it only assayed the binding of 3132 proteins, out of which 3037 were detected in the first measurement. After thresholding, 45 of these proteins turned out to be significant interaction partners of BIN1. Among the proteome that was detected in both measurements, 73 and 43 significant partners were identified in the first and second measurements, respectively. This included 27 partners that were significant in both measurements. However, 36 additional proteins show non-significant depletion in the first measurement that showed significant binding in the second and 10 additional proteins show non-significant depletion in the second measurement that showed significant binding in the first. If we consider the determined depletion values as true for these partners under the significance threshold,

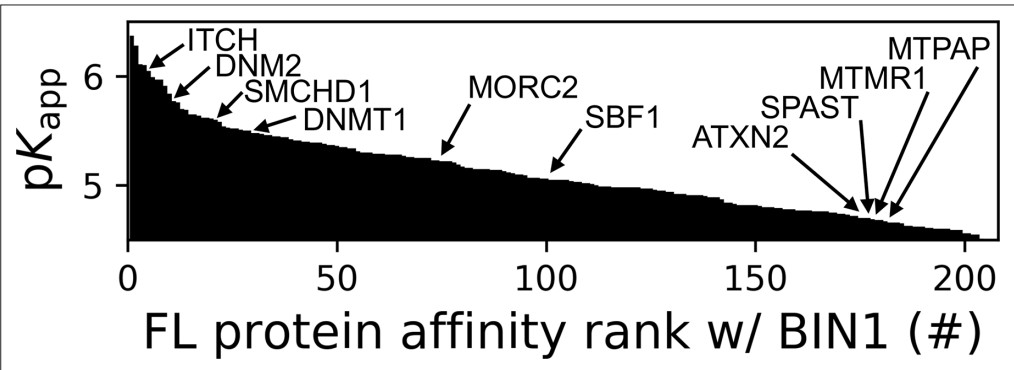

**Figure 4.** Affinity ranking of the 206 FL interaction partners of the BIN1 SH3 domain identified in nHU-MS experiments. Interaction partners found in previous studies, as well as partners whose importance was found to be significant in neuromuscular disorders are indicated. See ***Supplementary file 1*** for further details.

the two experiments showed a recall of 0.86, regardless of which measurement was considered as the point of reference (*Figure 3—figure supplement 3A*). Moreover, by comparing the affinities of the partners that we identified in these independent experiments, a strong proportionality was found with a statistical significant correlation. Finally, our analysis also revealed that high-affinity interactions were almost always found to be significant in both measurements, while weak interactions were only identified as significant in one of the two experiments. By combining the results of the two experiments, we assayed the binding of 6,453 FL endogenous proteins, out of which 206 showed significant binding to BIN1. The results of all affinity measurements of this work can be also accessed through the ProfAff affinity interactomic database, accessible at https://profaff.igbmc.science address.

To produce unbiased references for analyzing the BIN1_SH3 interactome, we carried out similar experiments for one related SH3 domain from Amphiphysin (AMPH) and four unrelated SH3 domains from Abelson murine leukemia viral oncogene homolog 1 (ABL1), Rho guanine nucleotide exchange factor 7 (ARHGEF7), Protein arginine N-methyltransferase 2 (PRMT2), and Obscurin (OBSCN). These other SH3 domains showed comparable promiscuity with BIN1 with the exception of OBSNC_SH3, whose interactome appears to be markedly less promiscuous with only a few detected interaction partners (*Figure 3—figure supplement 3*, *Supplementary file 1*). For these domains, we quantified pairwise interactomic similarities with BIN1_SH3 using the affinities of interaction partners that showed binding to both SH3 domains (*Figure 3*, *Figure 3—figure supplement 3*, *Supplementary file 1*). Based on affinities of the shared partners, AMPH_SH3 and ARHGEF7_SH3 show similar affinity profiles to BIN1_SH3 with statistically significant correlation. In contrast, the affinities of the shared partners between BIN1_SH3 and ABL1_SH3 or PRMT2_SH3 differ substantially with no significant correlation.

## BIN1 interacts with many proteins involved in neuromuscular disorders, besides DNM2

The 206 interaction partners of the SH3 domain of BIN1 identified in the nHU assay can be ranked based on their apparent affinity constants (*Figure 4*). Only three of these partners were previously found to bind BIN1 in high-throughput qualitative interactomic studies: DNM2, ITCH, and SMCHD1. These partners were found to rank among the strongest interaction partners of BIN1_SH3. Eight proteins (ATXN2, DNM2, DNMT1, MORC2, MTPAP, SBF1, SMCHD1, SPAST) are, like BIN1 itself, encoded by genes listed in the gene table of monogenic neuromuscular disorders (*Cohen et al., 2021*). Another significant BIN1_SH3 binder detected by our assay is MTMR1, a close paralog of myotubularin (MTM1) whose mutation can cause the X-linked form of CNM also called myotubular myopathy (*Laporte et al., 1996*; *Zanoteli et al., 2005*). Five out of these phenotypically-related partners (DNM2, SMCHD1, DNMT1, MORC2, and SBF1) were found to display relatively high affinity for BIN1_SH3 with a dissociation constant <10 μM (p$K_d$ >5), while ATXN2, SPAST, MTMR1, and MTPAP showed somewhat weaker affinities. Out of these, DNMT1, SPAST, and MTPAP only showed detectable binding to BIN1_SH3, while SMCHD1, MORC2, and MTMR1 also showed detectable binding to the SH3 domains of AMPH (*Supplementary file 1*). The remaining partners (DNM2, SBF1, ATXN2, ITCH) were found to be more promiscuous, displaying affinities for the SH3 domains of BIN1, AMPH, as well as other proteins involved in our screens. Nevertheless, pathological mutations of BIN1 that result a dysfunctional SH3 domain will have interactome-wide consequences, and such effects are not going to be restricted to its interaction with DNM2 but to its entire list of partners deciphered here.

## Deciphering the site-specific affinity interactome of the BIN1 SH3 using fragmentomic holdup

A common feature of 'co-complex' oriented approaches, such as nHU, is that we detect both direct and indirect interactions. To investigate the molecular mechanisms of the direct, site-specific interactions of the BIN1 SH3 domain, we used a bioinformatic-experimental combined approach. PRMs are extremely common in the human proteome. For example, only considering the two most common types of PxxP motifs, we could identify more than 10,000 motifs within ca. 5000 proteins in the ca. 20,000 proteins encoded in our genome (*Krystkowiak and Davey, 2017*; *Kumar et al., 2019*). Therefore, it is expected that 25% of the identified interaction partners of BIN1 will contain PRMs, even by chance. When we screened the sequences of all identified interaction partners looking for putative PRMs, we could identify such motifs in 65% of the interaction partners of BIN1 (133 out of 206),

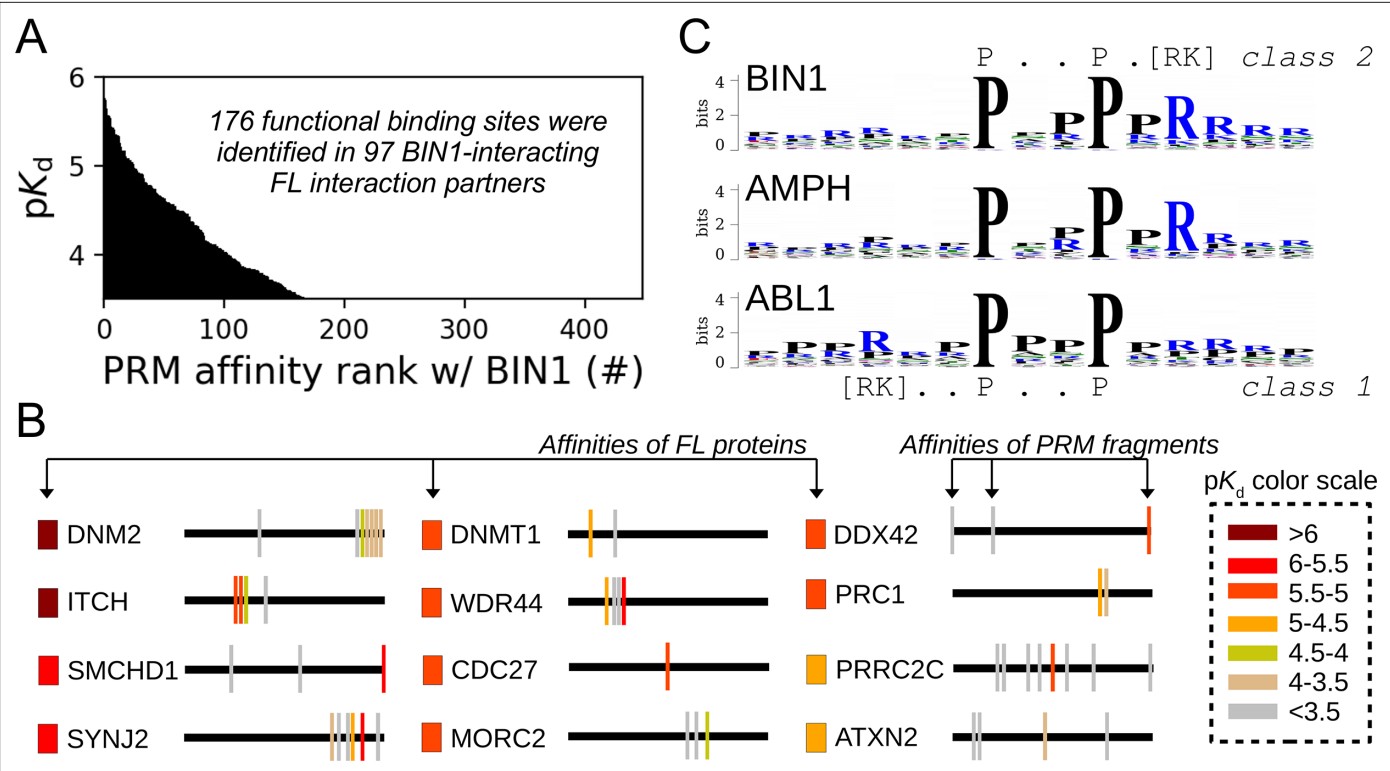

**Figure 5.** Affinity measurements between the SH3 domain of BIN1 and isolated PRM fragments. (**A**) Affinity profile of BIN1_SH3 measured using fragmentomic holdup against 448 synthetic PRMs found in FL interaction partners previously identified by nHU-MS. 176 PRMs were found to bind to BIN1 displaying affinities ranging from low micromolar to a few hundreds of micromolar dissociation constants. These motifs were found in 97 proteins, matching at least a single functional binding site for half of the identified FL interaction partners. (**B**) The combination of native and fragmentomic holdup reveals biophysical properties of FL proteins and elementary binding sites. The measured affinities of intact proteins are indicated with colored boxes and site-specific affinities of individual PRMs are indicated with colored spikes, where colors were adjusted to measured steady-state affinities of FL proteins and PRM sites, respectively. Note that the protein schemes are not to scale to the actual protein length, but are the approximate relative positions of indicated PRMs. (**C**) Affinity-weighted specificity logo of the SH3 domains of BIN1, AMPH, and ABL1. BIN1 and AMPH nearly uniquely interacts with class 2 PxxP motifs, while ABL1 prefers to bind class 1 PxxP motifs. See *Figure 5—figure supplement 1* and *Supplementary file 2* for further details.

The online version of this article includes the following source data and figure supplement(s) for figure 5:

**Figure supplement 1.** Site-specific affinity interactomes of other SH3 domains.

**Figure supplement 2.** Quality control of purified MBP-fused SH3 domains used for fragmentomic holdup experiments.

**Figure supplement 2—source data 1.** Original SDS-PAGE.

indicating a >2.5-fold enrichment compared to the random occurrence (*Figure 3*, *Supplementary file 1*). Within disordered regions of these proteins, we identified 417 putative PRMs matching the [RK]..P..P or P..P.[RK] consensus motifs (class 1 and class 2 PxxP motifs, respectively). We also identified 31 PRMs found in 19 potential interaction partners of BIN1 that showed ambiguous binding in the nHU experiments (strong binding with low statistical significance). Altogether, we identified 448 putative PRMs that may interact with BIN1 directly.

All putative PRMs were synthesized as 15-mer biotinylated peptides and their affinities were systematically measured against the SH3 domain of BIN1. Out of these, 176 motifs showed detectable binding with BIN1 (*Figure 5A*, *Supplementary file 2*). These BIN1-binding motifs were derived from 97 FL proteins, including 5 that showed ambiguous binding in nHU. Therefore, out of 133 interaction partners of BIN1_SH3 that contain putative PRMs, we matched 92 with at least one quantified site-specific affinity, annotating nearly half of all identified BIN1_SH3 partners as a likely direct interaction partner of BIN1. The remaining interaction partners that do not contain class 1 or class 2 PxxP motifs may interact with BIN1 indirectly, or via other types of PRMs belonging to other known or unknown motif classes. Surprisingly, we find a very poor correlation, with a PCC of 0.2, between measured

affinities of intact proteins and the affinities of isolated motifs, even if we only consider the best motif in each protein. Moreover, when we systematically compare these affinity differences, we find that >85% of partners bind stronger as an intact protein compared to any isolated PRM that we could identify in their sequence. These differences are often quite substantial, with a mean/median $pK_{app}$ (intact protein) - $pK_d$ (PRM) difference of 0.71/0.76, meaning that most isolated motifs bind at least five- to sixfold weaker than the full-length protein. Thus, interactions of SH3 domains are fairly atypical motif-mediated interactions and individual sites should be rather interpreted as parts of greater PRRs. This is consistent with the observations that partners of the Grb2 family SH3 proteins bind with higher affinities to larger regions of their ligands compared to short PxxP motifs, possibly due to the contribution of regions outside the core PxxP motifs in the binding mechanism (*Bartelt et al., 2015*).

The nHU assay alone does not indicate which part of the identified partner is responsible for the interaction. By complementing it with the fragmentomic holdup approach, we could not only decide which identified partners bind directly, but also determine the functional sites that contribute to the direct interactions, providing detailed mechanistic characterization for the 92 interactions that were found to directly interact with BIN1 (*Figure 5B*). For example, SMCHD1 mediates a high-affinity interaction with BIN1_SH3, but it contains three putative binding sites. Out of these, two turned out to be unable to recruit BIN1_SH3 in the holdup assay, while the third motif fragment mediates similarly strong affinity with BIN1_SH3 as FL SMCHD1. Some proteins contained more than one putative PRMs that bind BIN1_SH3 above detection. For example, instead of finding a single peptide that detectably interacts out of the many, we identified several PRMs within the PRR of DNM2 that all displayed weaker affinities in isolation than the previously measured affinity of the entire PRR of DNM2, or than full-length DNM2 (recombinant, or endogenous), indicating a high degree of synergism between the sites and a possible contribution of DNM2 oligomerization to the high affinity interaction with BIN1.

The site-specific PRM-binding profile of the BIN1 SH3 domain also provides a deeper insight into the PRM binding preferences of the SH3 domain itself. By comparing the PRM-binding affinities of the SH3 domain of BIN1 with the corresponding affinities of the five other SH3 domains addressed in our study, we have found that BIN1_SH3 has a clear preference for class 2 PxxP motifs over class 1 motifs, similarly to the SH3 domains of AMPH and ARHGEF7 (*Figure 5C*, *Figure 5—figure supplements 1–2*, *Supplementary file 2*). In contrast, PRMT2_SH3 does not appear to have a marked specificity and both ABL1_SH3 and OBSCN_SH3 appear to prefer class 1 PxxP motifs. In addition, the affinity profile of BIN1_SH3 was most similar to the affinity profile of AMPH_SH3, and was also similar to the affinity profile of ARHGEF7_SH3, but was more distinct from the affinity profiles of PRMT2_SH3 or ABL1_SH3. The SH3 domain of OBSCN was found to only mediate detectable binding to a handful of PRMs included in our panel signifying its peculiar nature compared to the other SH3 domains. Overall, these observations are in excellent agreement with previously observed biophysical similarities with the FL partner binding, indicating that the observed interactomic similarities and differences of these domains arise from the molecular nature of their interactions.

## The BIN1 SH3 interactome reveals the protein's critical role during cell cycle

The combination of our affinity interactomic approaches revealed a large set of previously unknown partners that appeared to interact with BIN1 in a direct manner, through at least one functional PxxP motif. We hypothesized that by analyzing all identified BIN1 partners, we could get a better view of the cellular mechanisms regulated by BIN1. We performed functional enrichment analysis to identify over-represented GO terms and carried out hierarchical clustering to identify groups of BIN1-partners that participate in related processes (*Figure 6A*, *Supplementary file 3*). This revealed that the identified partners of the SH3 domain of BIN1 are most often involved in two types of biological processes, either related to nuclear processes (such as DNA replication or mRNA processing) or to mitotic processes. We complemented this clustering with an evolutionary scoring that measured the degree of conservation of the identified functional sites. For this, orthologous sequences were collected and we measured how far the presence of BIN1-binding PxxP motifs can be traced back in evolution. Based on this, we could identify multiple deeply conserved BIN1-interaction motifs in proteins involved in multiple biological processes. Most interestingly, although the binding site of DNM2 was found to be extremely conserved, it was not the most conserved site as the interaction site found in PRC1 was possible to trace back to opisthokonts. PRC1 was originally identified as a partner

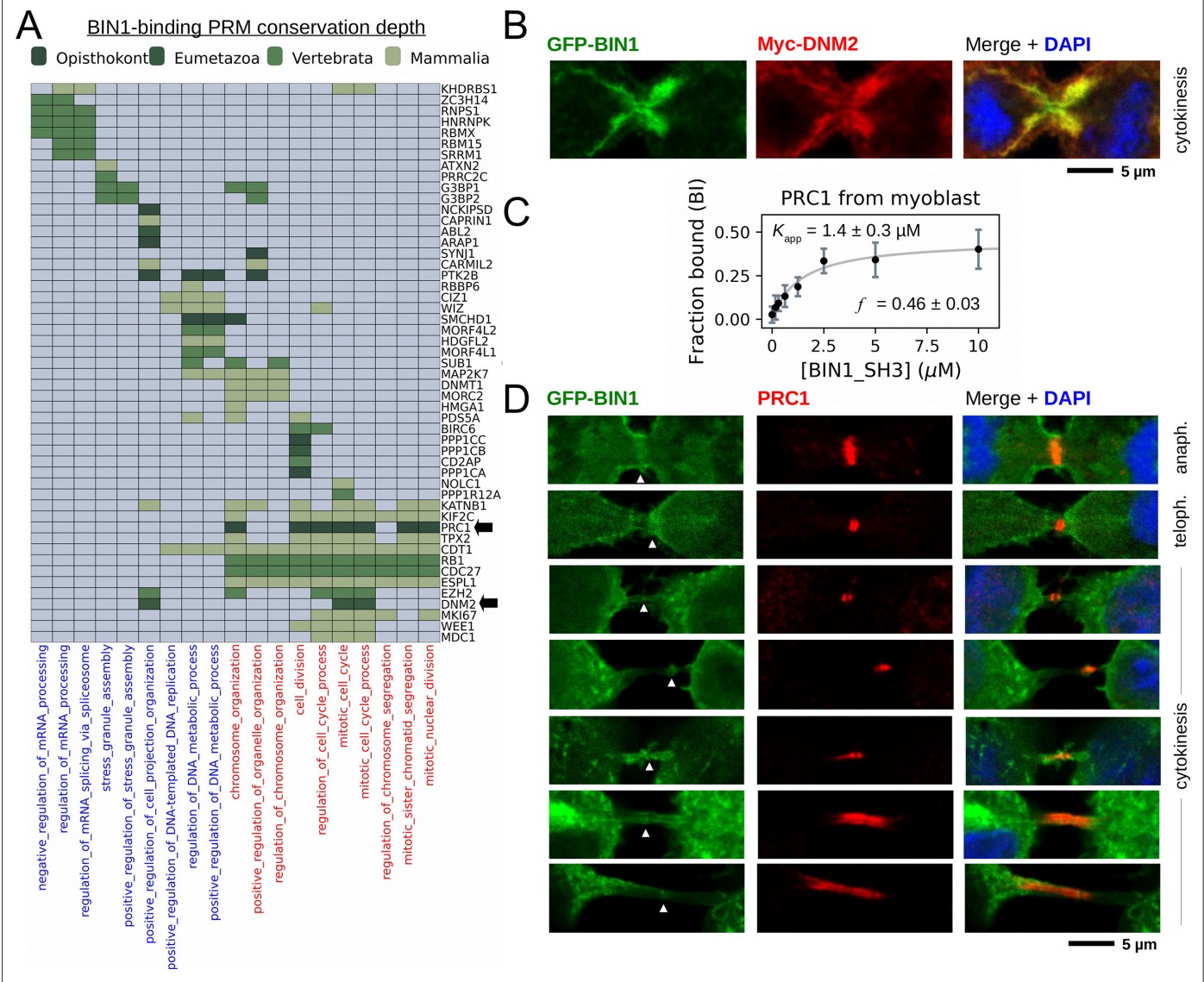

**Figure 6.** BIN1 interacts with multiple proteins involved in the mitotic phase and is localized at the membrane bridge formed between the daughter cells. (**A**) Functional clustering of the identified BIN1 partners that contains BIN1-binding PxxP motifs. At the bottom of the panel, nuclear, or nucleic-acid-related processes are colored in blue and mitotic processes are colored in red. Heatmap color coding is according to the conservation depth of the highest affinity BIN1-interacting motif. See **Supplementary file 3** for data. (**B**) During cytokinesis, BIN1 and DNM2 were found to localize at the cleavage furrow. A representative image of dividing Cos-1 cells, that were transfected with GFP-BIN1 and Myc-DNM2. (**C**) Titration nHU to further characterize the interactions of BIN1 with PRC1. PRC1 binding is measured from the same binding experiment using myoblast extract, that was used to characterize DNM2 binding (**Figure 3A**). (The average and standard deviation of three nHU western blot experiments are shown.) (**D**) Representative confocal images of the membrane bridge between daughter cells. Cos-1 cells were transfected with GFP-BIN1 and stained for endogenous PRC1. White arrows indicate the cleavage furrow or the midbody. For additional supporting confocal images see **Figure 6—figure supplement 1**.

The online version of this article includes the following figure supplement(s) for figure 6:

**Figure supplement 1.** Cellular translocation of BIN1 during mitosis and cytokinesis.

of the SH3 domain of BIN1 in our proteome-wide nHU interactomic screen displaying a medium affinity. Later, we identified two potential PRMs in its sequence that both displayed binding activity to the SH3 domain of BIN1 with comparable affinities to the full-length PRC1 (**Figure 5**). Interestingly, these tandem PRMs of PRC1 overlap with mitotic phosphorylation sites and a previously characterized nuclear localization signal that regulates PRC1 localization during the cell cycle (**Jiang et al., 1998**).

Our findings suggest that BIN1 has a so far unknown role in mitotic processes. To further investigate this mechanism, we re-analyzed our membrane tubulation assays and searched for dividing cells that

were transfected with BIN1 (WT) and DNM2 (*Figure 6B*). Although we had difficulties capturing some rare mitotic stages in these transfected cells, we have found that it was easier to find dividing cells using the F584S BIN1 variant that behaved similarly to WT regarding the observed changes during mitosis. Surprisingly, all dividing cells (WT or F584S) either turned out to be completely devoid of BIN1 tubular structures, or the short membrane tubules were restricted to the cell periphery (*Figure 6—figure supplement 1*). In late anaphase, and in the early stages of cytokinesis, the membrane tubules reappeared in cells and were found in higher density around the cleavage furrow and the midbody. Thus, membrane structures caused by BIN1 displayed dramatic rearrangement during the mitotic phase.

In addition to DNM2, many other partners of BIN1 were also connected to mitotic processes, including PRC1. To further investigate this interaction in muscle-related context in a more quantitative manner, we probed our previous nHU titration experiment with myoblast extract using an antibody against endogenous PRC1 (*Figure 6C*). This experiment confirmed this interaction and we have found that PRC1 displayed a moderate affinity with an apparent dissociation constant of 1.4 µM (p$K$app = 5.86), which is approximately >10-fold weaker than the BIN1-DNM2 interaction. Although PRC1 is known to localize at the nucleus during interphase, it translocates to the mitotic spindle midzone during anaphase and at the cleavage furrow and the midbody during cytokinesis (*Mollinari et al., 2002*). Since we observed that BIN1 localizes at the same sites in the DNM2 co-localization experiments, we decided to investigate whether a possible cellular encounter may exist at these sites between PRC1 and BIN1 and we transfected Cos-1 cells with GFP-BIN1 (WT or F584S) and stained them for endogenous PRC1 (*Figure 6D*, *Figure 6—figure supplement 1*). Similarly to the previous experiment, in cells transfected with only BIN1 (and at endogenous DNM2 level), the BIN1-induced membrane tubules also showed dramatic rearrangement during mitosis, when they either completely decondensed or were restricted to the cell periphery. As expected, in interphasic cells BIN1 and PRC1 are well separated as PRC1 is only found in the nucleus and BIN1 remains in the cytoplasm at membrane tubules. Yet, once cells entered the mitotic phase, PRC1 localizes to the same cellular regions as BIN1 during anaphase and cytokinesis, that is cleavage furrow and midbody (*Figure 6D*, *Figure 6—figure supplement 1*). Further investigation is needed to definitively establish the direct association between PRC1 and BIN1 at these sites since this local enrichment can be also an indirect consequence of interactions mediated by other proteins in this area.

## The impact of missense variants on the affinity interactome of BIN1

The created synthetic PRM motif library, which comprises all putative class 1 or class 2 PxxP motifs that we could identify in the FL interaction partners of BIN1 provides a near exhaustive picture of cognate BIN1 interaction sites that we could find in the human proteome. We used this resource to measure the site-specific affinity interactomes of the eight natural BIN1 variants addressed above. After measuring the affinity interactomes of the BIN1_SH3 variants, we compared their affinity profiles to the WT BIN1_SH3 by calculating cumulative Euclidean affinity distances (*Figure 7*, *Figure 5—figure supplement 2*, *Supplementary file 2*). Note that we chose Euclidean distances, because $\Delta$p$K_d$ is proportional to $\Delta\Delta$G, hence the calculated Euclidean distance quantifies the overall differences in binding energy differences of multidimensional affinity spaces. We found that only those variants caused substantial perturbation in the affinity interactome of BIN1 that we also identified as a perturbing variants with our conventional screening (*Figure 2*). Both Y531S, D537V cause perturbed affinity profiles (PAP) with a general affinity interactome reshuffling, in which most peptide targets mediate weaker affinities compared to the WT SH3 domain. In contrast, the mutation F584S causes a near complete loss of function (LOF). Besides, based on the very similar interactomic properties of the other variants to WT BIN1, it is likely that these are benign variants. In support of this, the common variant T532M, that displays affinity interactome quasi identical to WT BIN1, is present 116 times at homozygous state in genomic databases with no connection to any clinical phenotype (*Karczewski et al., 2020*).

From a structural perspective, Y531S, T532M, D537V, and Q540H are situated on the so-called RT loop, P551L can be found on the n-Src loop, while V566M, V583I, R581C, and F584S are integral part of the β-strands of the SH3 fold (*Figure 7B*). The Y531S and D537V variants, which created an affinity profile reshuffling, are placed on the RT loop and are integral parts of the PRM binding interface. In contrast, the F584S variant, which abolished most interactions, is placed on the β4-strand facing the hydrophobic core. Thus, it is likely that the Y531S and D537V variants impact the PRM binding in a

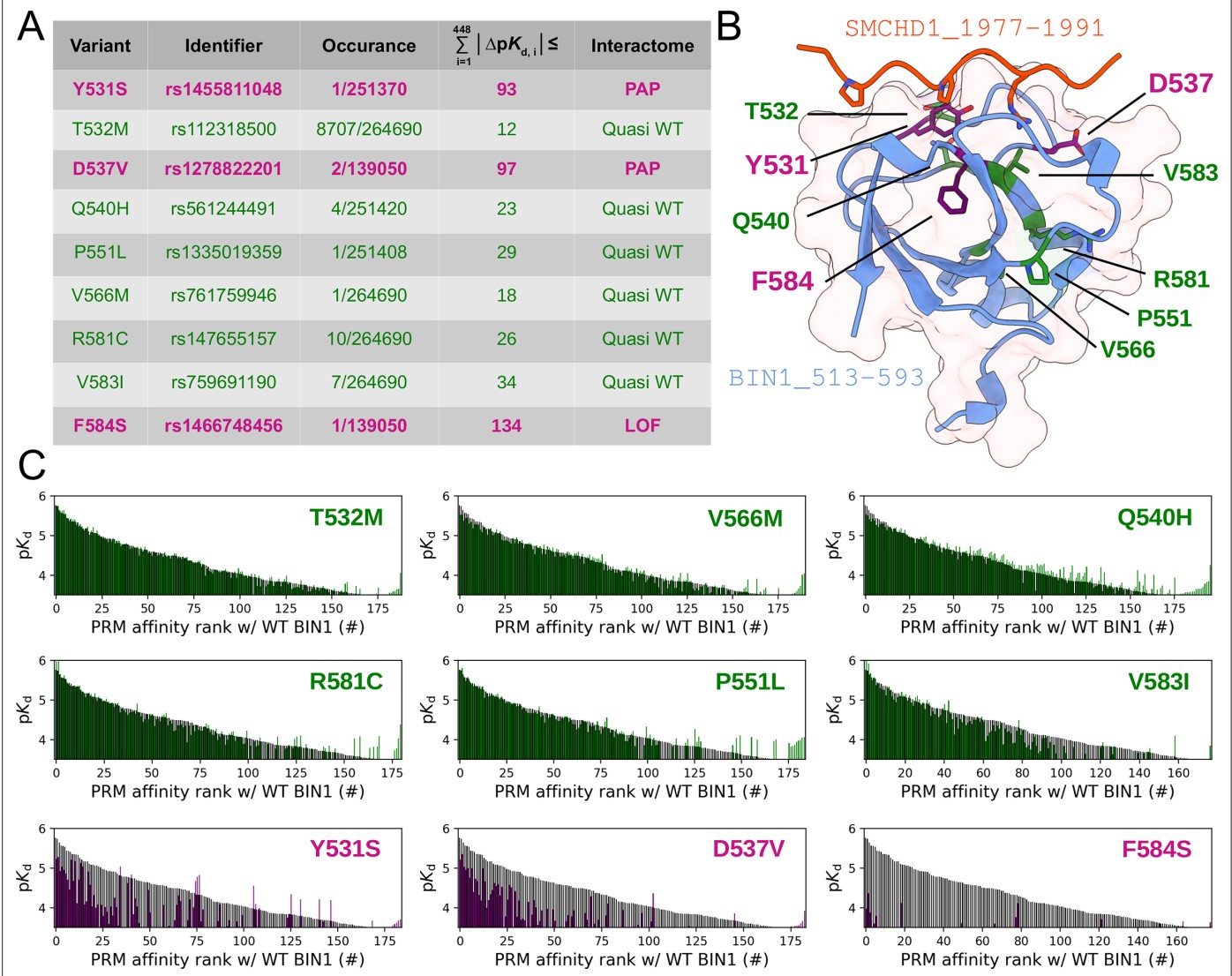

**Figure 7.** Natural variants of BIN1 can cause affinity perturbation at an interactomic scale. (**A**) A summary of our affinity interactomic tests performed with 9 natural variants of the BIN1 SH3 domain. The cumulative Euclidean affinity distances to the WT BIN1, calculated based on the explored 448 dimensional affinity space, are indicated for each variant. For affinities where no detectable binding was observed the detection threshold was used for calculation, hence only the lower limit of the Euclidean distance could be estimated. Variants colored green have minor effect on affinity interactomes, while variants colored in purple displaying either perturbed affinity profiles (PAP) or general loss of functions (LOF). (**B**) Location of the assayed variants on the structure of WT BIN1. D537 and Y531 are positioned near the PRM binding interface, F584 is buried in the hydrophobic core of the SH3 domain. The structure of the BIN1 SH3 domain bound to a high affinity peptide taken from SMCHD1 was generated using AlphaFold2 (***Tunyasuvunakool et al., 2021***). (**C**) Affinity interactome profiles of the BIN1 variants (colored in green or purple) compared with the affinity profile of WT BIN1 (colored in black). Motifs in the affinity profiles were ranked according to their affinities measured with WT BIN1. Only the motifs displaying detectable binding out of the 449 assayed PRMs are shown. See ***Supplementary file 2*** for further details.

The online version of this article includes the following figure supplement(s) for figure 7:

**Figure supplement 1.** Predicting the structural consequences of the likely pathogenic variants of BIN1 with Alphafold2.

direct manner and the F584S variant causes a destabilized SH3 fold. Since this variant was found to be capable of recruiting DNM2 in cellular assays, it is likely that this *in vitro* destabilization can be partially rescued in the context of full-length BIN1, possibly by intramolecular interactions with other regions of the protein (***Kojima et al., 2004***; ***Wu and Baumgart, 2014***).

In light of the recent advancement in structure prediction, we assessed if AlphaFold2 Multimer is capable of predicting the structural consequences of these potential pathological variants. We used the AlphaFold2-Multimer in ColabFold v1.5.2 to predict the structures of complexes of the likely

pathological variants bound to the high-affinity PRM found in SMCHD1 that show the optimal binding sequence for BIN1 (*Mirdita et al., 2022*; *Tunyasuvunakool et al., 2021*). The resulting models all predicted complexes that made structural sense with no obvious indication of their perturbed affinity interactomes (*Figure 7—figure supplement 1*). In the case of the variants Y531S and D537V, only very minor local changes could be detected in the conformation of the bound peptide and only in a fraction of the predicted models. In the case of the F584S variant, nothing indicated either decreased stability of the domain or perturbations in the bound conformation of the motif. Thus, we concluded that standard modern structure-prediction tools are not yet fully capable of foreseeing the consequences of sequence variants. Therefore, experimental approaches, such as the holdup assays are still going to be essential to accurately measure the consequences of missense variants. Furthermore, predicting interactomic affinity-reshuffling that we observed with some of the BIN1 SH3 domain variants would not only require the precise and faithful prediction of bound complexes with all partners but also the prediction of changes in binding energies that singular structural snapshots cannot easily provide.

## Discussion

Interactions mediated by short linear motifs are highly transient and routinely used interactomic techniques often fail to detect them (*Kassa et al., 2023*). Consequently, past studies identified only a handful of BIN1 interaction partners (*Figure 1*). In the present work, we used a state-of-the-art affinity interactomic approach taking full advantages of the recently developed native holdup and the fragment-based holdup techniques. We identified ~200 interaction partners of the SH3 domain of BIN1. We also identified SH3 domain binding motifs within half of them that were able to interact with BIN1 with comparable affinities to the full-length proteins. By analyzing these partners, we revealed a potentially critical role of BIN1 in the cell cycle. We have found that membrane tubules formed by BIN1 display dramatic rearrangement during cell division that includes a nearly complete decondensation and a rapid reassembly during telophase. We have also found that BIN1 localizes to the cleavage furrow during telophase and at the midbody during cytokinesis. This connection between BIN1 and mitotic processes is surprising, yet not entirely unexpected. Both proteomic and transcriptomic data showed that BIN1 expression fluctuates throughout the cell cycle similarly to other regulators of the cell cycle, albeit to a more modest extent (*Santos et al., 2015*). It has been also showed that DNM2 co-localizes with microtubule bundles formed at the midbody (*Thompson et al., 2002*). During cytokinesis, PRC1, a new partner of BIN1 identified here, is known to cross-link microtubule bundles at the midbody to promote the division of the daughter cells (*Mollinari et al., 2002*). Our study demonstrates, that BIN1 is also involved in this process, where it is likely that the membrane remodeling activities of the BAR domain of BIN1 is mediating regulatory functions. Our holistic approach also revealed that BIN1 participates in the regulation of the mitotic phase by interacting with several proteins connected to these processes, including not only DNM2 but also PRC1 and many others. To unveil the precise mechanism of how BIN1 participates in the mitotic phase, further investigation will be necessary.

Noteworthy, mutations in BIN1 or DNM2 lead to different forms of CNM. We could confirm that DNM2 ranks among the highest affinity interactions of BIN1, signifying the importance of this interaction. However, we also identified many other, previously unknown partners of BIN1 that displayed similar affinities as DNM2 and that may be also critical for understanding the molecular mechanisms through which mutations in BIN1 contribute to the development of CNM. For example, we identified SMCHD1 as a strong interaction partner of BIN1, displaying a similar affinity as DNM2. Mutations of this protein are also linked to a neuromuscular disorder called facioscapulohumeral muscular dystrophy (FSHD) and both FSHD and CNM patients suffer from muscle weakness as a major symptom (*Cohen et al., 2021*). Our unbiased study suggests that their complex may play a role in the symptomatic manifestations of both conditions. Nevertheless, even if the impact of the newly discovered BIN1 interaction partners will turn out to be somewhat less critical than DNM2 in CNM, disruptive mutations in the SH3 domain of BIN1 will necessarily perturb all the identified interactions that collectively contribute to CNM.

Rare variants of uncertain clinical significance present a major challenge in interpreting genetic results. Affinity interactomic approaches allow the investigation of the consequences of naturally occurring variants in a highly quantitative manner (*Weimer et al., 2023*). Even in a small protein

region, such as the SH3 domain of BIN1, dozens of sequence variations can be found in genomic databases, most of which are associated with unknown clinical significance. Using our approach, we could demonstrate that most of these variants do not cause detectable perturbation in the intrinsic affinity interactome of BIN1, but we could also identify three rare variants that caused affinity rewiring, leading to altered molecular phenotypes related to BIN1. Although genetic approaches could not determine statistically significant causality between these mutations and the pathology due to their sparsity, our affinity interactomic approach has associated them with putative clinical risk. In conclusion, we demonstrated that affinity interactomics is not limited to the identification and characterization of interaction partners, but is also suitable for testing effects of sequence variations in order to identify and validate potentially disease-causing mutations.

## Materials and methods

### Cloning and protein expression, purification

SH3 domain coding sequences (BIN1, UniProt ID O00499, residues 513–593; AMPH, UniProt ID P49418, residues 615–695; PRMT2, UniProt ID P55345, residues 24–96; OBSCN, UniProt ID Q5VST9, residues 5594–5674; ARHGEF7, UniProt ID Q14155, residues 178–251; ABL1, UniProt ID P00519, residues 56–121) were obtained from cDNA pools using standard protocols. For nHU and fragmentomic holdup reactions, SH3 domains were cloned as His$_6$-AviTag-MBP-TEV-SH3, or His$_6$-MBP-TEV-SH3 constructs in custom pET vectors, respectively. The empty His$_6$-AviTag-MBP-TEV-STOP vector was used to produce biotinylated MBP for nHU control experiments. BIN1 variants were created with standard QuickChange strategy.

Biotinylated proteins were co-expressed with BirA (PET21a-BirA, Addgene #20857) in *E. coli* BL21(DE3) cells. At Isopropyl β-D-1-thiogalactopyranoside (IPTG) induction (1 mM IPTG at 18 °C for ON), 50 μM biotin was added to the media. Harvested cells were lysed in a buffer containing 50 mM TRIS pH 7.5, 150–300 mM NaCl, 50 μM biotin, 2 mM 2-mercaptoethanol (BME), cOmplete EDTA-free protease inhibitor cocktail (Roche, Basel, Switzerland), 1% Triton X-100, and trace amount of DNAse, RNAse, and Lysozyme. Lysates were frozen at –20 °C before further purification steps. Lysates were sonicated and centrifuged for clarification. Expressed proteins were captured on pre-packed Ni-IDA (Protino Ni-IDA Resin, Macherey-Nagel, Duren, Germany) columns, were washed with at least 10 column volume cold wash buffer (50 mM TRIS pH 7.5, 150 mM NaCl, 2 mM BME) before elution with 250 mM imidazole. The Ni-elution was collected directly on a pre-equilibrated amylose column (amylose high flow resin, New England Biolabs, Ipswich, Massachusetts). Amylose column was washed with 5 column volume cold wash buffer before fractionated elution in a buffer containing 25 mM Hepes pH 7.5, 150 mM NaCl, 1 mM TCEP, 10% glycerol, 5 mM maltose, cOmplete EDTA-free protease inhibitor cocktail. The concentration of proteins was determined by their UV absorption at 280 nm before aliquots were snap frozen in liquid nitrogen and were stored at –80 °C. Non-biotinylated proteins were produced identically but without co-transformation with BirA and without supplementing the media or the lysis buffer with biotin. As a quality control, the double-affinity purified His$_6$-MBP-fused SH3 domains were loaded on SDS-PAGE and stained with Coomassie brilliant blue (*Figure 5—figure supplement 2*).

### Purification and enzymatic characterization of FL DNM2

Human DNM2 protein was produced from pVL1392 plasmid in Sf9 cells with the baculovirus system as described previously (*Lionello et al., 2022*). Briefly, a transfection was performed with the DNM2 plasmid to produce viruses. Sf9 cells were infected with viruses and grown for 3 d at 27 °C and then centrifuged at 2000 x *g* for 10 min. DNM2 recombinant protein was resuspended in buffer A (20 mM HEPES, pH 7.4; 150 mM NaCl, 5% of Glycerol, 1 mM EGTA, 1 mM DTT) and purified with GST-BIN1_SH3 bound to Glutathione-Sepharose 4B beads (GE Healthcare). Human SH3 of BIN1 with GST tag (GST-SH3) was produced from pGEX6P1 plasmid in *Escherichia coli* BL21. *E. coli* producing this recombinant protein were induced with 1 mM IPTG for 3 hr at 37 °C, centrifuged at 7500 x *g*, and the protein was purified using Glutathione Sepharose 4B beads (GSH-resin). The BIN1_SH3-bound DNM2 was eluted with buffer B (20 mM PIPES, pH 6,8; 1200 mM NaCl). After elution, the pooled elution fractions were dialyzed with buffer A (*Figure 3—figure supplement 1A*).

GTPase activity of recombinant DNM2 was measured with malachite green assay as previously described with a reaction time of 10, 30 or 180 min at 37 °C (*Gómez-Oca et al., 2022*; *Figure 3— figure supplement 1B*). DNM2 recombinant protein was incubated with 2-Diacyl-sn-glycero-3-phospho-L-serine (PS, 4 µg/ml) and 30 mM of NaCl. The concentration of GTP in the reaction mix was 0.3 mM. The tested concentrations of DNM2 recombinant protein were from 2 to 64 nM.

## Peptide synthesis

The DNM2 PRR peptide (residues 823–860) was chemically synthesized on an ABI 443 A synthesizer with standard Fmoc strategy with biotin group attached to the N-terminus via a TTDS (Trioxatridecan-succinamic acid) linker and was HPLC purified (>95% purity). Predicted peptide mass was confirmed by MS and peptide concentration was determined based on dry weight.

The PxxP peptide library was prepared as described in details before (*Gogl et al., 2022*). Briefly, peptides were synthetized with standard Fmoc strategy in 96-well plate format using an Intavis multipep Rsi. Peptides were amidated C-terminally and were N-terminally tagged with biotin via an Ado-Ado (Ado = 8-amino-3,6-dioxaoctanoic acid), or a Glu-Glu-Ado-Ado linker. Predicted peptide masses were confirmed by MS and average peptide concentrations were determined based on the excess weight of the entire 96-well plate after drying and were used in 10–50×molar excess during the holdup experiments.

## Mammalian cell extract preparation

Jurkat E6.1 cells (ECACC #88042803, RRID: CVCL_0367) were grown in RPMI (Gibco) medium completed with 10% FBS (Gibco BRL) and 40 µg/ml gentamicin (Gibco/Life Technology). The C25 myoblast cell line obtained from Institut de Myologie (Paris, France) were grown below 60% confluency in DMEM/199 medium (Sigma-Aldrich) supplemented with 20% FBS, 25 µg/ml fetuin (Sigma-Aldrich), 0.5 ng/ml basic fibroblast growth factor (Gibco BRL), 5 ng/ml epidermal growth factor (Gibco BRL), 0.2 µg/ml dexamethasone (Sigma-Aldrich), 5 µg/ml insulin (Eli Lilly Co., Indianapolis, USA), 50 U/ml penicillin (Gibco/Life Technology), and 100 µg/ml gentamicin. All cells were kept at 37 °C and 5% $CO_2$. To prepare total cell extracts, Jurkat cells were seeded onto T-175 flasks and grew until $3x10^6$ cells/ml confluency, C25 myoblasts were seeded on T-75 flasks and grew until they reach ½ confluency, where we detected the highest expression for DNM2 in these cells before. Jurkat cells were collected by 1000 *g* x 5 min centrifugation, washed with PBS and then lysed in ice-cold lysis buffer (Hepes-KOH pH 7.5 50 mM, NaCl 150 mM, Triton X-100 1%, cOmplete EDTA-free protease inhibitor cocktail 5 x, EDTA 2 mM, TCEP 5 mM, glycerol 10%). C25 myoblasts were lysed with the same lysis buffer directly on the flasks after washing them with PBS, and the cells were collected by scraping. Lysates were sonicated 4x20 s with 1 s long pulses on ice, then incubated rotating at 4 °C for 30 min. The lysates were centrifuged at 12,000 rpm 4 °C for 20 min and supernatant was kept. Total protein concentration was measured by standard Bradford method (Bio-Rad Protein Assay Dye Reagent #5000006) using a BSA calibration curve (MP Biomedicals #160069, diluted in lysis buffer) on a Bio-Rad SmartSpec 3000 spectrophotometer instrument. Lysates were diluted to 2 mg/ml concentration, aliquoted and snap-frozen in liquid nitrogen and stored at –80 °C until measurement.

## Single-point nHU experiment

For single-point nHU experiments carried out at ~10 µM estimated bait concentration, pre-equilibrated 25 µl streptavidin resin (Streptavidin Sepharose High Performance, Cytiva) was incubated with 1 ml 25–40 µM purified biotinylated MBP or MBP-fused SH3 domains for 1 hr at room temperature. After the incubation, all resins were washed with 20 resin volume (500 µl) holdup buffer (50 mM Tris pH 7.5, 300 mM NaCl, 1 mM TCEP, .22 filtered). The washed resins were then mixed with 25 µl 1 mM biotin solution, diluted in 10 resin volume holdup buffer and were incubated for 10 min at room temperature. Then, the resulting resins were washed three more times with 20 resin volume holdup buffer. The resulting SH3-saturated resins were mixed with 100 µl 2 mg/ml Jurkat extracts and were incubated at 4 °C for 2 hr with constant mild agitation. After the incubation ended, the solid and liquid phases were separated by a brief centrifugation (15 s, 2000 × *g*) and 70 µl of the supernatant was recovered rapidly. Then, to minimize carryover contamination from resin, the recovered supernatants were centrifuged one more time and 50 µl of the supernatant was recovered that was subjected for mass spectrometry analysis. As described in details before, measurements were done in singlicates

against duplicate controls with injection triplicates during MS measurements (*Zambo et al., 2022*). The reason to use injection triplicates instead of experimental triplicates is to get as accurate prey depletion as possible from the mass spectrometry measurements as these measurements are typically less robust compared to the actual nHU experiments. The experiment series carried out with the 6 SH3 domains was analyzed on the Orbitrap Exploris 480 MS and the measurement with BIN1_SH3 alone was analyzed with Orbitrap Elite.

MS analysis was performed as described in details before (*Zambo et al., 2022*). Briefly, nHU samples were precipitated with TCA, and the urea-solubilized, reduced and alkylated proteins were digested with trypsin and Lys-C at 2 M final urea concentration. Peptide mixtures were then desalted on C18 spin-column and dried on Speed-Vacuum. 100 ng peptide mixtures were analyzed using an Ultimate 3000 nano-RSLC coupled in line, via a nano-electrospray ionization source, with a LTQ-Orbitrap ELITE mass spectrometer (Thermo Fisher Scientific, San Jose, California) or with the Orbitrap Exploris 480 mass-spectrometer (Thermo Fisher Scientific, Bremen, Germany) equipped with a FAIMS (high Field Asymmetric Ion Mobility Spectrometry) module. Data was collected in DDA (data dependent acquisition) mode, proteins were identified by database searching using SequestHT (Thermo Fisher Scientific) with Proteome Discoverer software (Thermo Fisher Scientific). Peptides and proteins were filtered with a false discovery rate (FDR) at 1%. Label-free quantification was based on the extracted ion chromatography intensity of the peptides. All samples were measured in technical triplicates. The measured extracted ion chromatogram (XIC) intensities were normalized based on median intensities of the entire dataset to correct minor loading differences. For statistical tests and enrichment calculations, not detectable intensity values were treated with an imputation method, where the missing values were replaced by random values similar to the 10% of the lowest intensity values present in the entire dataset. Unpaired two tailed T-test, assuming equal variance, were performed on obtained $\log_2$ XIC intensities. All raw LC-MS/MS data have been deposited to the ProteomeXchange via the PRIDE database with identifier PXD040169.

Obtained fold-change values were converted to apparent affinities using the hyperbolic binding equation and binding thresholds were determined as described before (*Zambo et al., 2022*). Proteins containing PRMs were identified with the help of SliMSearch using the class 1 and class 2 PxxP consensus motif definitions found in the ELM database (LIG_SH3_1 and LIG_SH3_2; *Krystkowiak and Davey, 2017*; *Kumar et al., 2019*).

## Titration nHU and HU experiments

Titration holdup experiments were carried out as described above using 25 µl saturated resins prepared (*Zambo et al., 2022*). Briefly, we mixed MBP, or BIN1_SH3 saturated resins and certain proportions and kept the total resin-analyte ratio constant during the experiment (for 25 µl we used 100 µl analyte). Experiments were carried out at 4 °C for 2 hr and recovered supernatants were subjected to Western blot. As analyte, either total myoblast extracts (2 mg/ml) were used in the case of titration nHU experiments, or 62 nM purified DNM2 in the case of titration HU experiments.

Samples were mixed with 4 x Laemmli buffer (120 mM Tris-HCl pH 7, 8% SDS, 100 mM DTT, 32% glycerol, 0.004% bromphenol blue, 1% β-mercaptoethanol) in 3:1 ratio. Equal amounts of samples were loaded on 10% acrylamide-gels. Transfer was done into PVDF membranes using a Trans-Blot Turbo Transfer System and Trans-Blot Turbo RTA Transfer Kit (BioRad, #1704273). After 1 hr of blocking in 5% milk, membranes were incubated overnight 4 °C in primary DNM2 antibody (1:1000, in-house antibody #2641, rabbit polyclonal, IGBMC) in 5% milk or in primary PRC1 antibody (1:1000, Sigma-Aldrich #HPA034521, rabbit polyclonal, RRID: AB_10670169) in 5% milk. Membranes were washed three times with TBS-Tween and incubated at RT for 1 hr in secondary antibody (goat anti-rabbit(H+L) #111-035-003 RRID: AB_2313567) in 5% milk (dilution: 1:10,000). After washing three times with TBS-Tween, membranes were exposed to chemiluminescent HRP substrate (Immobilon, #WBKLS0100) and captured in docking system (Amersham Imager 600, GE). Then, membranes were exposed to 15% $H_2O_2$ to remove secondary signal and the membranes were incubated with anti-GAPDH primary antibody (1:5000, Sigma #MAB374, clone 6C5, RRID: AB_2107445) for 1 hr at room temperature. After three washings, the membranes were incubated with the secondary antibody (goat anti-mouse(H+L) #115-035-146 RRID: AB_2307392) in 5% milk (concentration 1:10,000), washed three times and captured in the docking system as above. Densitometry was carried out on raw Tif images by using Fiji ImageJ 1.53 c.

## Fragmentomic holdup assay

Fragmentomic holdup assays were carried out in 384 well filter plates using intrinsic fluorescence as a readout following the exact same protocol that was described in details before (*Gogl et al., 2022*). Briefly, 5 µl of streptavidin resin, pre-saturated with peptides, were aliquoted on filter plates and the holdup reaction was carried out with 10 µl analyte in holdup buffer, complemented with 4 µM double-affinity purified MBP-fused SH3 domain, as well as 50 nM fluorescein and 100 nM mCherry as internal standards. Filtrates were analyzed on a PHERAstar (BMG Labtech, Offenburg, Germany) microplate reader by using 485 ± 10nm–528 ± 10nm (fluorescein), 575 ± 10nm–620 ± 10nm (mCherry), and 295 ± 10nm–350 ± 10nm (Trp-fluorescence) band-pass filters. Filter plates with peptide-saturated beads were recycled as before. However, we have found that unlike PDZ-binding motifs, PxxP motifs were difficult to recycle several times and an apparent decrease of bait concentration was often found, which was possible to minimize by long incubations in holdup buffer. We hypothesize that this phenomena is caused by some sort of hydrophobic collapse. Regardless, we decided to only recycle each filter plate only a few times (5–10, while we could safely recycle PDZ-binding motif saturated plates >20 times). When small apparent bait concentration decrease was obtained, we corrected the measured values based on previous measurements. In the case of BIN1 variants that were found to show perturbed interactomes, measurements were repeated on fresh filter plates to eliminate the possibility of disturbing the results (e.g. false negatives or positives). Affinity-weighted specificity logos were calculated as described before (*Gogl et al., 2022*). The obtained affinities (of both peptide motifs and of FL proteins obtained) were uploaded to the ProfAff affinity database and accessible at https://profaff.igbmc.science (*Gogl et al., 2022*).

## Membrane tubulation assay

pTL1 myc-His plasmids containing the human DNM2 and pEGFP-BIN1 plasmid (human BIN1 isoform 8) have been previously used (*Nicot et al., 2007*). Mutant versions of BIN1 were created by standard QuickChange mutagenesis protocol.

Cos-1 cells (ATCC #CRL-1650, RRID: CVCL_0223) were grown in DMEM (1 g/L glucose) containing 10% FCS and 40 µg/mL gentamicin, kept at 37 °C and 5% $CO_2$ and were split twice a week for maintaining. The cells were tested negative for mycoplasma prior to the experiments. The day before transfection, $0.375 \times 10^5$ cells were seeded in the wells of a 24-well plate with coverslips. Cells were co-transfected with 0.5 µg of DNM2 and 0.25 µg of BIN1 (either WT or mutants) per well using JetPRIME reagent (Polyplus, #101000046) according to the manufacturer's recommendations. For single transfection experiments, cells were transfected with 0.25 µg of BIN1 (either WT or F584S mutant) per well using JetPRIME reagent (Polyplus, #101000046) according to the manufacturer's recommendations. The medium was changed to fresh medium after 5 hr of transfection to enhance survival after transfection.

Immunostaining was carried out after 24 hr of transfection. Cells were washed once with sterile PBS and fixed with 4% formaldehyde solution for 15 min at room temperature. After washing three times with PBS, cells were permeabilized with 0.2% Triton X-100 in PBS for 10 min. Then, cells were blocked in blocking solution (30 mL PBS, 1.5 g BSA powder (MP Biomedicals #160069), 0.1% Triton X-100) for 1 hr at room temperature. Cells were incubated with the primary antibody anti-myc (Thermo Fisher, clone 9E10, #13–2500, RRID: AB_86583, dilution: 1:500) or anti-PRC1 (Sigma-Aldrich #HPA034521, RRID: AB_10670169, dilution: 1:200) in blocking solution overnight at 4 °C. The next day, cells were washed three times with PBS and were incubated with secondary antibody Alexa Fluor 594 conjugated anti-mouse (Invitrogen, #A-11032, RRID: AB_2534091, dilution: 1:1000) or anti-rabbit (Invitrogen, #A-11037, RRID: AB_2534095, dilution: 1:1000) in blocking solution for 1 hr at room temperature. Cells were washed again three times with PBS, and coverslips were mounted with DAPI containing Vectashield (#H-1200) on slides. Images were taken using a Leica SP5 confocal microscope (Leica Camera AG, Wetzlar, Germany) with an HCX PL APO 63×/1.40–0.60 oil objective using excitation at 405 nm (diode), 488 nm (Argon laser), and 594 nm (HeNe laser) and emission at 415–480, 510–560, and 610–695 nm for DAPI (nucleus), GFP (BIN1), and Alexa 594 (DNM2 or PRC1), respectively. Image analysis was done using Fiji ImageJ 1.53 c software.

To determine single cell co-localization of BIN1 variants and DNM2, Pearson correlation coefficients were calculated using Coloc 2 plugin with auto-threshold in ImageJ. In every image, those cells were only selected by ROI, which showed membrane tubules in the green (BIN1) channel and

expressed both GFP-BIN1 and DNM2, i.e. there were signal in both green and red channels for the given cell. Statistics were done using GraphPad Prism 7 software.

## Ortholog database

To compile evolutionary data for each protein containing BIN1 binding motif, we created a dataset of orthologous sequences. These sequences were obtained by running the GOPHER prediction algorithm against the UniProt Reference proteome database with default settings (*Davey et al., 2007*; *UniProt Consortium, 2023*). Subsequently, we performed multiple sequence alignments of the orthologs for each protein using the MAFFT algorithm with default parameters (*Katoh et al., 2002*). To classify the ortholog sequences, we utilized the UniProt taxonomic lineage, employing the five main evolutionary levels, Mammalia, Vertebrata, Eumetazoa, and Unicellular (only eukaryotic), to determine the most specific term for each sequence. Then, protein level conservation of each BIN1 partner was defined based on the orthologs with the most distantly related taxonomic term. For the evolutionary analysis, a minimum of three predicted orthologs was necessary at each level.

## Position specific scoring matrix (PSSM)

To generate a BIN1 binding motif specific PSSM, 175 measured motifs were used. The motifs were applied for PSSM constructing as 15 long regions in which from position 7–10 were the consensus PxxP residues. The elements of the PSSM ($P_{i,j}$) were expressed as the log-odds score of amino acid frequency in each position in the known motifs divided by the background frequency. As not every amino acid was present in each position in the known set, a pseudo-count correction was introduced:

$$P_{i,j} = log\left(\frac{A_{i,j} + \frac{B}{20}}{\frac{m+B}{D_j}}\right) \tag{1}$$

where $A_{i,j}$ is the frequency of amino acid i at position j in the known motif set, and $D_i$ is the background frequency of amino acid i. The background frequency was calculated using the eukaryotic proteome from UniProt (*UniProt Consortium, 2023*). B is the pseudocount with a value of 5 (*Nishida et al., 2009*), m is the number of sequences, and 20 is the number of amino acids.

## Motif conservation

Based on the multiple sequence alignments of orthologs, we analyzed each aligned instance of the 175 BIN1 binding motifs in terms of PSSM-based conservation. The PSSM score for each orthologous motif was calculated and then normalized using the human motif PSSM score as a reference. Subsequently, for each taxonomic level, we computed an average conservation score based on the normalized PSSM scores of the orthologous motifs belonging to that level. The calculated conservation scores of evolutionary levels were then used to determine motif conservation. A BIN1 binding motif was considered conserved at a given level if the evolutionary level score exceeded 0.5.

## Functional enrichment

GO enrichment analysis was carried out by the g:Profiler tool with default parameter setting (*Raudvere et al., 2019*). Overrepresentation test of GO terms was applied for 98 BIN1 partners containing the 175 motifs used in the evolutionary analysis. From the enriched terms, non-specific, generic terms (more than 1000 annotations) have been discarded. A total of 47 significantly enriched terms were obtained, 5, 24 and 18 Molecular Function, Biological Process and Cellular Component GO aspects, respectively.

## Hierarchical clustering and heatmap

For clustering, we gathered and preprocessed the protein data along with their associated GO-term annotations. This dataset consisted of a binary matrix, where each row represented a protein and each column corresponded to a specific GO term. The matrix cells were filled with binary values (0 or 1) indicating the presence or absence of a given GO term annotation for a particular protein. Only Biological Process GO terms (24) were used in our clustering procedure. Next, the Seaborn heatmap

python library in conjunction with hierarchical clustering algorithms was used to create protein clusters and visually represents them (*Waskom, 2021*). For refinement, redundant clusters in terms of GO terms were deleted, retaining 19 out of 24 clusters. The heatmap showcases the protein clusters as well-defined blocks, where each row corresponds to a protein and each column corresponds to a GO term.

## Acknowledgements

We thank Pascal Eberling for the synthesis of the DNM2 PRR peptide, Camille Kostmann for help in cloning, Nadege Diedhiou for her help in the tubulation assay, as well as the cell culture and the photonic microscopy platform of the IGBMC for their help in cell culturing and imaging. GG was supported by the collaborative post-doc grant of IGBMC and Inserm. The project was supported by the Ligue contre le cancer (équipe labellisée 2015 to GT), the Agence Nationale de la Recherche (grant ANR-18-CE92-0017, ANR-22-CE44-0018, and ANR-22-CE11-0026 to GT), AFM-Téléthon (23933), and OTKA (K139284 to ZD). As a member of the IGBMC institute, we benefited from the French Infrastructure for Integrated Structural Biology (FRISBI) ANR-10-INSB-05–01, from Instruct-ERIC, from IdEx Unistra (ANR-10-IDEX-0002), from SFRI-STRAT'US project (ANR 20-SFRI-0012), and from EUR IMCBio (ANR-17-EURE-0023) under the framework of the French Investments for the Future Program as a member of the Interdisciplinary Thematic Institute IMCBio, as part of the ITI 2021–2028 program of the University of Strasbourg, CNRS and Inserm.

## Additional information

### Funding

| Funder | Grant reference number | Author |
| --- | --- | --- |
| Institut de génétique et de biologie moléculaire et cellulaire | post doc grant | Gergo Gogl |
| Inserm | CRCN starting package | Gergo Gogl |
| Ligue Contre le Cancer | 2015 | Gilles Trave |
| Agence Nationale de la Recherche | ANR-18-CE92-0017 | Gilles Trave |
| Agence Nationale de la Recherche | ANR-22-CE44-0018 | Gilles Trave |
| Agence Nationale de la Recherche | ANR-22-CE11-0026 | Gilles Trave |
| AFM-Téléthon | 23933 | Jocelyn Laporte |
| OTKA | K139284 | Zsuzsanna Dosztanyi |

The funders had no role in study design, data collection and interpretation, or the decision to submit the work for publication.

### Author contributions

Boglarka Zambo, Conceptualization, Data curation, Formal analysis, Investigation, Visualization, Methodology, Writing – review and editing; Evelina Edelweiss, Bastien Morlet, Matyas Pajkos, Zsuzsanna Dosztanyi, Formal analysis, Investigation, Methodology, Writing – review and editing; Luc Negroni, Investigation, Methodology, Writing – review and editing; Soren Ostergaard, Resources, Formal analysis, Investigation, Methodology, Writing – review and editing; Gilles Trave, Supervision, Funding acquisition, Writing – review and editing; Jocelyn Laporte, Conceptualization, Funding acquisition, Writing – review and editing; Gergo Gogl, Conceptualization, Resources, Data curation, Formal analysis, Supervision, Funding acquisition, Validation, Investigation, Visualization, Methodology, Writing - original draft, Writing – review and editing

## Author ORCIDs

Jocelyn Laporte ⬤ https://orcid.org/0000-0001-8256-5862
Gergo Gogl ⬤ http://orcid.org/0000-0002-8597-3711

Reviewer #1 (Public Review): https://doi.org/10.7554/eLife.95397.4.sa1
Reviewer #2 (Public Review): https://doi.org/10.7554/eLife.95397.4.sa2
Author response https://doi.org/10.7554/eLife.95397.4.sa3

## Additional files

### Supplementary files

• Supplementary file 1. Results of the nHU-MS experiments. Each sheet contains the results of a single experiment performed with a given bait. Significant partners have numerical values in the "pKdsign" column.

• Supplementary file 2. Results of the fragmentomic holdup experiments. Significant partners have non "0" values in the "pKd" column.

• Supplementary file 3. Results of the bioinformatic analysis. The first sheet contains the evolutionary conservation of PxxP motifs. The second sheet contains the results of the GO enrichment analysis.

• MDAR checklist

### Data availability

All data needed to evaluate the conclusions in the paper are present in the paper, in the Supplementary Materials and in public databases. Raw mass spectrometry data are available in the PRIDE database with identifiers PXD040169.

The following dataset was generated:

| Author(s) | Year | Dataset title | Dataset URL | Database and Identifier |
|---|---|---|---|---|
| Negroni L, Gogl G | 2024 | Affinity-ranking BIN1 interactions and screening natural BIN1 variants that may cause neuromuscular disorders | https://www.ebi.ac.uk/pride/archive/projects/PXD040169 | PRIDE, PXD040169 |

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
