## [Editor Report · eLife assessment]

This work describes a novel affinity interactomics approach that allows investigators to identify networks of protein-protein interactions in cells. The **important** findings presented here describe the application of this technique to the SH3 domain of the membrane remodeling Bridging Integrator 1 (BIN1), the truncation of which leads to centronuclear myopathy. The authors present **solid** evidence that BIN1 SH3 engages with an unexpectedly high number of cellular proteins, many of which are linked to skeletal muscle disease, and evidence is presented to suggest that BIN1 may play a role in mitosis creating the potential for new avenues in drug development efforts. Some of the findings, however, remain rather preliminary, lack sufficient replicates and may require additional experiments to definitively support the conclusions.

---

## [Referee Report · Reviewer #1 (Public Review)]

Summary:

In this paper, Zambo and coworkers use a powerful technique, called native holdup, to measure the affinity of the SH3 domain of BIN1 for cellular partners. Using this assay, they combine data using cellular proteins and proline-containing fragments in these proteins to identify 97 distinct direct binding partners of BIN1. They also compare the binding interactome of the BIN1 SH3 domain to the interactome of several other SH3 domains, showing varying levels of promiscuity among SH3 domains. The authors then use pathway analysis of BIN1 binding partners to show that BIN1 may be involved in mitosis. Finally, the authors examine the impact of clinically relevant mutations of the BIN1 SH3 domain on the cellular interactome. The authors were able to compare the interactome of several different SH3 domains and provide novel insight into the cellular function of BIN1. Generally, the data supports the conclusions, although the reliance on one technique and the low number of replicates in each experiment is a weakness of the study.

Strengths:

The major strength of this paper is the use of holdup and native holdup assays to measure the affinity of SH3 domains to cellular partners. The use of both assays using cell-derived proteins and peptides derived from identified binding partners allows the authors to better identify direct binding partners. This assay has some complexity but does hold the possibility of being used to measure the affinity of the cellular interactome of other proteins and protein domains. Beyond the utility of the technique, this study also provides significant insight into the cellular function of BIN1. The authors have strong evidence that BIN1 might have an undiscovered function in cellular mitosis, which potentially highlights BIN1 as a drug target. Finally, the study provides outstanding data on the cellular binding properties and partners of seven distinct SH3 domains, showing surprising differences in the promiscuity of these proteins.

Weaknesses:

There are several weaknesses of the study. First, the authors rely completely on a single technique to measure the affinity of the cellular interactome. The native holdup is a relatively new technique that is powerful yet relatively unproven. However, it appears to have the capacity to measure the relative affinity of proteins and the authors describe the usefulness of the technique. Second, and most important, the authors use a relatively small number of replicates for the holdup assays. The holdup technique will have biological variation in the cellular lysate or purified protein that could impact the results, so more replicates would enhance the reliability of the results.

---

## [Referee Report · Reviewer #2 (Public Review)]

Summary:

The authors report here interesting data on the interactions mediated by the SH3 domain of BIN1 that expand our knowledge on the role of the SH3 domain of BIN1 in terms of mediating specific interactions with a potentially high number of proteins and how variants in this region alter or prevent these protein-protein interactions. These data provide useful information that will certainly help to further dissect the networks of proteins that are altered in some human myopathies as well as the mechanisms that govern the correct physiological activity of muscle cells.

Strengths:

The work is mostly based on improved biochemical techniques to measure protein-protein interaction and provide solid evidence that the SH3 domain of BIN1 can establish an unexpectedly high number of interactions with at least a hundred cellular proteins, among which the authors underline the presence of other proteins known to be causative of skeletal muscle diseases and not known to interact with BIN1. This represents an unexpected and interesting finding relevant to better define the network of interactions established among different proteins that, if altered, can lead to muscle disease. An interesting contribution is also the detailed identification of the specific sites, namely the Proline-Rich Motifs (PRMs) that in the interacting proteins mediate binding to the BIN1 SH3 domain.

Weaknesses:

Less convincing, or too preliminary in my opinion, are the data supporting BIN1 co-localization with PRC1. Indeed, the affinity of PRC1 is significantly lower than that of DNM2, an established BIN1 interacting protein. Thus, this does not provide compelling evidence to support PRC1 as a significant interactor of BIN1. Similarly, the localization data appears somewhat preliminary to substantiate a role of BIN1 in mitotic processes. These findings may necessitate additional experimental work to be more convincing.

---

## [Author Response]

The following is the authors’ response to the previous reviews.

**Reviewer #1**

We modified the text regarding PRC1 according to the reviewer’s recommendation.

**Reviewer #2**

Following the reveiwer’s advise, we introduced the holdup assay, as well as the native holdup assay in more details.

This new part now also discusses the question of replicates in more details. We do not agree with the eLife assessment on this matter, but we think that this assessment was made because analyzing holdup data requires a different approach compared to more conventional interactomic approaches and these differences were not introduced in sufficient depth. We hope that the inclusion of more background reasoning, as well as by providing a more detailed comparison of the measured independent BIN1 interactomes, now included on Figure S4, will eliminate all confusion in the reader.

We thank the reviewer for guiding us to a previous work that was done on Grb2. Indeed, the finding of this earlier work aligns perfectly with our finding suggesting general similarities in SH3 domain mediated interactions.